# Multilineage-Differentiating Stress-Enduring Cells (Muse Cells): The Future of Human and Veterinary Regenerative Medicine

**DOI:** 10.3390/biomedicines11020636

**Published:** 2023-02-20

**Authors:** María Gemma Velasco, Katy Satué, Deborah Chicharro, Emma Martins, Marta Torres-Torrillas, Pau Peláez, Laura Miguel-Pastor, Ayla Del Romero, Elena Damiá, Belén Cuervo, José María Carrillo, Ramón Cugat, Joaquín Jesús Sopena, Mónica Rubio

**Affiliations:** 1Bioregenerative Medicine and Applied Surgery Research Group, Department of Animal Medicine and Surgery, CEU Cardenal Herrera University, CEU Universities, C/Tirant lo Blanc, 7, Alfara del Patriarca, 46115 Valencia, Spain; 2Garcia Cugat Foundation CEU-UCH Chair of Medicine and Regenerative Surgery, 08006 Barcelona, Spain

**Keywords:** muse cells, multilineage-differentiating stress-enduring cells, mesenchymal stem cells, stem cells, regenerative medicine

## Abstract

In recent years, several studies have been conducted on Muse cells mainly due to their pluripotency, high tolerance to stress, self-renewal capacity, ability to repair DNA damage and not being tumoral. Additionally, since these stem cells can be isolated from different tissues in the adult organism, obtaining them is not considered an ethical problem, providing an advantage over embryonic stem cells. Regarding their therapeutic potential, few studies have reported clinical applications in the treatment of different diseases, such as aortic aneurysm and chondral injuries in the mouse or acute myocardial infarction in the swine, rabbit, sheep and in humans. This review aims to describe the characterization of Muse cells, show their biological characteristics, explain the differences between Muse cells and mesenchymal stem cells, and present their contribution to the treatment of some diseases.

## 1. Introduction

Mesenchymal stem cells (MSCs) are somatic stem cells that can be easily collected from mesenchymal tissues, such as bone marrow (BMSCs) or adipose tissue (AMSCs). Due to their differentiation ability and immunomodulatory function, MSCs have been used in the treatment of many diseases in different species [1], such as osteoarthritis in humans [2,3] and rabbits [4], ischemic stroke in humans [5,6] and Alzheimer’s disease in humans [7,8] and a mouse model [9]. These cells have been defined as multipotent cells because of their potential to differentiate into several mesenchymal lineages, including bone marrow (BM), adipose tissue, bone, cartilage, tendon or muscle [10]. However, in many cases, this ability was demonstrated in a heterogeneous population of MSCs, leaving the question of whether MSCs really were multipotent stem cells [11].

In 2010, a subpopulation of stem cells hidden within BMSCs were discovered by Dezawa’s research group. These cells were called multilineage-differentiating stress-enduring (Muse) cells due to their unique properties [11]. Muse cells are endogenous, pluripotent, nontumorigenic, self-renewing and stress-tolerant stem cells. Additionally, they can be obtained from various tissues of the adult organism, such as fat, skin or bone marrow, avoiding the ethical problem of embryonic stem (ES) cells [12,13,14,15,16]. Moreover, they have anti-apoptotic, anti-fibrosis and tissue-protective effects at the damage site. Muse cells repair the tissue and replace damaged or apoptotic cells by spontaneous differentiation into tissue-constituent cells [17]. For all these advantages, human Muse cells have been considered the future of regenerative medicine [18].

In humans, Muse cells can be effectively isolated from BM and connective tissues as they are positive for CD105 and stage-specific embryonic antigen-3 (SSEA-3) markers [11,19,20]. The first marker is common to MSCs and the second one is a human pluripotent stem cell marker that can also be found in the undifferentiated state of human ES cells [19,20]. Since their discovery, several studies have highlighted all the specific characteristics of these cells and their application in the treatment of different diseases, such as stroke or acute myocardial infarction (AMI) [21,22], among others.

This review describes the characterization of human Muse cells and equivalent cells found in some animal species, shows the main characteristics of these cells, compares them with those present in MSCs and, finally, explains the progress in the treatment of some diseases using Muse cells.

## 2. Characterization of Muse Cells

Muse cells and muse-like cells have been characterized in different species since their discovery. The main data revealed by various studies are summarized in Table 1.

### 2.1. Human

The proportion of Muse cells (described as SSEA-3+ cells) within BMSCs and fibroblasts has been found to account for 1.1% ± 0.05% and 1.8% ± 0.22%, respectively. Muse cells obtained from fibroblasts were reported to spend 1.3 days per cell division. After day 8, they decreased in proliferation until day 14. Furthermore, it was determined that 11.6% ± 1.6% of fibroblasts Muse-enriched cell populations (MEC) and 8.1% ± 0.2% of BMSCs MEC are able to form Muse-cell-derived cell clusters (M-clusters). By day 14, when cells stopped dividing, the M-clusters had a maximum size of 150 μm [11].

After 7 days in gelatin-coated dishes, immunohistochemistry showed that some of the cells of the M-clusters were positive for different markers of the ectoderm (neurofilament-M), mesoderm (α-smooth muscle actin, desmin) and endoderm (α-fetoprotein, cytokeratin 7), and RT-PCR confirmed that they indeed expressed markers of all three germ layers [11].

Moreover, these cells expressed the pluripotency genes OCT3/4, NANOG and SOx2, and M-clusters were positive to alkaline phosphatase (ALP) staining [11].

Muse cells can be isolated using the SSEA-3 marker from different tissues including bone marrow, adipose tissue and peripheral blood, among others [11,23,24,25,26,27]. However, markers differ between different locations. Human Muse cells obtained from bone marrow and connective tissues are known to be SSEA-3+ and CD105+ [11,23], as explained before, whereas Muse cells isolated from peripheral blood are SSEA-3+ and positive for the white cell marker CD45 [28]. Another study has also reported that Muse cells obtained from human skin tissue were CD29+ and CD90+, which are mesenchymal markers [23].

In human BM, cells double positive for SSEA-3 and CD105 markers (designed as Muse cells) account for approximately 0.04% of mononucleated cells [11]. It has been suggested that these cells are constantly mobilizing directly from the BM to the peripheral blood, where they also represent 0.04% of mononuclear cells [28]. Muse cells are believed to arrive to every organ through the bloodstream, explaining why their distribution is so wide [29]. Nevertheless, as markers differ between BM and peripheral blood, it remains unknown if Muse cells change classes while they are mobilizing or if there are two lines of Muse cells working at the same time [28].

Finally, returning to the isolation of Muse cells, it has been reported that Muse cells can be isolated directly from tissue samples as SSEA-3/CD105 double-positive cells by fluorescence-activated cell sorting (FACS) [11,23]. However, isolation as SSEA-3+ cells has been reported to be successful when the basal population is composed of cultured mesenchymal cells (e.g., BMSCs) since nearly all cells will be positive for mesenchymal markers such as CD105 [15].

### 2.2. Rabbit

The presence of Muse cells within the BMSCs has been studied and described, and they account for approximately 0.5% of their population, counting SSEA-3+ cells as Muse cells. By cell sorting, Muse cells (SSEA-3+) were separated from non-Muse cells (SSEA-3). Only Muse cells formed M-clusters. Both Muse cells and M-clusters expressed the pluripotency genes OCT3/4, NANOG and REx-1, although levels were higher in M-clusters [30].

When M-clusters were transferred to gelatin-coated dishes, cells started to express α-fetoprotein (endodermal marker, 2–3%), smooth muscle actin (mesodermal marker, 6%) and Tuj1 (ectodermal marker, 2–3%) after 2 weeks in culture, indicating that Muse cells can differentiate into cells of the three germ layers without induction [30].

Moreover, this study also confirmed the ability of rabbit Muse cells to spontaneously differentiate into cardiac-lineage cells, as they expressed GATA-4 and Nkx2.5 after 2 weeks on gelatin culture. Rabbit Muse cells were also cultured in cardiac differentiation induction medium, and after 10 days, 51% expressed cardiac troponin-I and 22% expressed sarcomeric α-actinin [30].

### 2.3. Mouse

Mouse Muse cells were isolated and characterized from BMSCs (BMSC-Muse cells), AMSCs from subcutaneous fat (AMSC-Muse cells) and fibroblasts from ear connective tissue (FIB-Muse cells). The number of SSEA-3+ cells (designated as Muse cells) was 3% of BM-MSCs, 2.6% of AMSCs and 2.1% of fibroblasts [14].

After 10 days in vitro, the M-clusters reached a size of 48 ± 17 μm for BMSC-Muse cells, 67 ± 12 μm for AMSC-Muse cells and 98 ± 15 μm for FIB-Muse cells [14].

Regarding the expression of stem cell markers, 75% of BMSC-Muse cells and 75% of FIB-Muse cells expressed the markers OCT3/4 and SOx2, whereas only 50% of FIB-Muse cells and 70% of BMSC-Muse cells expressed the NANOG marker. On the other hand, most of the AMSC-Muse cells were positive for OCT3/4, and only some of them expressed SOx2 and NANOG markers. Additionally, the mRNA expression of pluripotency genes was evaluated, demonstrating that BMSC-Muse cells had higher expression of SOx2 and NANOG than AMSC-Muse cells and FIB-Muse cells, and OCT3/4 mRNA expression was slightly decreased in these two groups than in BMSC-Muse cells [14].

Finally, the spontaneous differentiation ability was evaluated after 14 days in vitro in gelatin-coated dishes. The three populations of Muse cells expressed markers of three germ layers, including neurofilament light (NF-L, ectodermal marker), desmin (mesodermal marker) and CK-7 (endodermal marker). These results were obtained with immunocytochemical analysis, but RT-qPCR showed similar results. Moreover, it was also discovered that FIB-Muse cells showed a higher tendency to ectodermal commitment rather than mesodermal and endodermal lineage [14].

### 2.4. Rat

Rat Muse cells were isolated from BMSCs by 8-h trypsin-incubation stress and then cultured in a single-cell culture to form M-clusters. Approximately 2.03% ± 0.14% of total BMSCs formed M-clusters [31].

Muse cells underwent flow cytometry analysis and revealed that these cells were CD29+, CD90+ and RT1A+ while they were negative for CD34, CD45 and RT1B. Additionally, 75.6% ± 2.8% of Muse cells were SSEA-3+ and 74.8% ± 3.1% were SSEA-1+, whereas only 2.3% ± 0.3% and 2.1% ± 0.2% of BMSCs were positive for both, respectively. Furthermore, 77.62% ± 5.3% of SSEA-1+ cells were also SSEA-3+ [31].

Rat Muse cells expressed pluripotent stem cell markers such as NANOG, OCT3/4 and SOx2, detected with immunofluorescence assay. Their capacity to differentiate into cells of the three germ layers by induction was demonstrated, as they finally expressed α-fetoprotein (endodermal marker), α-smooth muscle actin (mesodermal marker) and neurofilament medium polypeptide (ectodermal marker) [31].

### 2.5. Swine

Swine SSEA-3+ cells (designated as Muse cells) represented 2% of swine BMSCs. As described in human Muse cells, swine Muse cells formed M-clusters with a diameter of 50 μm, like ES cell-derived embryoid bodies [32].

Swine Muse cells showed a higher expression of pluripotency genes, including POU5F1, SOx2 and NANOG, than non-Muse cells of BMSCs [32].

Additionally, it is suggested that swine Muse cells can differentiate into cells of the three germ layers, because when M-clusters were transferred to a gelatin-coated adherent culture for 10–14 days, endodermal, mesodermal and ectodermal markers were found [32].

### 2.6. Dog

The isolation of a cell subpopulation within canine AMSCs capable of surviving 16 h of long-term trypsin incubation (LTT) has been described (LTT-tolerant cells). These cells were not described as Muse cells but show similar characteristics. LTT-tolerant cells formed cell clusters in suspension culture of approximately 50 μm in diameter, like the embryoid body. The percentage of LTT-tolerant cells was 9.3% ± 0.5% of canine AMSCs [33].

LTT-tolerant cells were CD44+ and CD90+. Moreover, these cells expressed pluripotency markers such as SSEA-1, SSEA-3, NANOG, SOx2, OCT3/4 and TRA-1–60. The expression intensity of SSEA-1 and the expression of *FUT9* mRNA was reported to be higher after LTT, whereas the expression intensity of SSEA-3+ remained the same [33].

Regarding their differentiation ability, LTT-tolerant cells differentiated in vitro into cells of the three germ layers when they were induced. More studies about the spontaneous differentiation in vitro of LTT-tolerant cells are needed [33].

Finally, LTT-tolerant cells showed other similar properties to human Muse cells, such as self-renewal ability or being positive for ALP activity [33].

### 2.7. Goat

SSEA-3+ stem cells have been found within skin fibroblasts. As for cells found in the dog, these cells were not described as Muse cells but showed some similar properties. SSEA-3+ cells formed clusters (larger than 25 μm by day 4) positive for SSEA-3, CD105, NANOG, OCT4, and SOx2, TRA-1–60 and negative for SSEA-1 and SSEA-4. Additionally, they expressed the pluripotent genes OCT4, SOx2 and NANOG [34].

These cells were stress-tolerant, as they survived LTT. The population of SSEA-3+ cells accounted for 13% when goat fibroblasts were subjected to LTT, whereas only 3–4% of SSEA-3+ cells were found in fibroblasts without LTT [34].

The ability to differentiate to cells from the three germ layers was demonstrated both in vitro and in vivo [34].

Nevertheless, an important property of human Muse cells was not present in goat SSEA-3+ cells, showing a key difference with the first ones. Muse cells are known to be nontumorigenic, but these pluripotent cells found within goat fibroblasts formed teratomas in vivo when they were injected in mice [34]. Therefore, goat SSEA-3+ cells are not as useful as human Muse cells in cell therapy due to their tumorigenic potential. More studies are needed to confirm this important difference with human Muse cells.

### 2.8. Sheep

Muse cells were obtained from the adipose tissue and its quantity accounted for 1.4 × 10^5^ cells/cm^3^. These cells were SSEA-3+, CD166+, CD44+ and CD45-. The expression of SSEA-3+ was greater in Muse cells than in AMSCs [35].

Additionally, Muse cells presented significantly higher gene expression of the markers OCT3/4 and NANOG compared with AMSCs [35].

Finally, the expression of the angiogenic genes vascular endothelial growth factor (VEGF), angiogenin (ANG) and placental growth factor (PGF) was measured, and a significantly greater expression was detected in Muse cells than in the AMSCs [35].

**Table 1 biomedicines-11-00636-t001:** Characterization of Muse cells and equivalent cells in different species.

Study	Specie	Cell	Number of Cells	Triploblastic Differentiation	Expression of Pluripotency Markers and Genes	Expression of Other Markers
Kuroda et al., 2010 Wakao et al., 2011 Sato et al., 2020 [11,23,28]	Human	Muse cells	1.1% ± 0.05% of BMSCs 1.8% ± 0.22% of fibroblasts 0.04% of mononuclear cells (BM and PB)	Yes	SSEA-3, NANOG, OCT3/4, SOX2	CD105 (in Muse cells from BM and connective tissues). CD45 (in Muse cells from PB)
Yamada et al., 2018 [30]	Rabbit	Muse cells	0.5% of BMSCs	Yes	SSEA-3, NANOG, OCT3/4, REX-1	
Aprile et al., 2021 [14]	Mouse	Muse cells	3% of BM-MSCs 2.6% of AMSCs 2.1% of fibroblasts	Yes	SSEA-3, NANOG, OCT3/4, SOX2	
Sun et al., 2020 [31]	Rat	Muse cells	2.03% of total BMSCs (formed M-clusters)	Yes	RT1A, SSEA-1, SSEA-3, NANOG, OCT3/4, SOX2	CD29, CD90
Iseki et al., 2021 [32]	Swine	Muse cells	2% of BMSCs	Yes	SSEA-3, NANOG, SOX2, POU5F1	
Mitani et al., 2021 [33]	Dog	LTT-tolerant cells	9.3% of AMSCs	Yes	SSEA-1, SSEA-3, NANOG, OCT3/4, SOX2, TRA-1-60	CD44, CD90
Yang et al., 2013 [34]	Goat	SSEA-3+ stem cells	3–4% of fibroblasts	Yes	SSEA-3, NANOG, OCT4, SOX2, TRA-1-60	CD105
Castillo et al., 2023 [35]	Sheep	Muse cells	1.4 × 10^5^ cells/cm^3^	Yes	SSEA-3, NANOG, OCT3/4	CD44, CD166

BMSCs: bone marrow-derived mesenchymal stem cells; BM: bone marrow; PB: peripheral blood; AMSCs: adipose-derived mesenchymal stem cells; M-clusters: Muse-cell-derived cell clusters.

## 3. Biological Characteristics of Muse Cells

As described above, Muse cells are endogenous pluripotent stem cells that were demonstrated to have spontaneous differentiation ability into cells of ectodermal, endodermal, and mesodermal lineages; self-renewability ability; high tolerance to stress; and reparative function in vivo; and they are nontumorigenic [11,23,25,26].

### 3.1. Pluripotency

A major characteristic of Muse cells is their pluripotency. Muse cells have been shown to be capable of self-renewal, as well as differentiation into cells of ectodermal, mesodermal and endodermal lineages, both in vitro and in vivo [11].

Some studies have reported in vitro differentiation of Muse cells into the three germ layers by induction, for example into keratinocytes, melanocytes, cardiomyocytes or neuronal cells, among others [23,36,37,38]. The rate of this induced differentiation accounts for 80–95% [29].

To prove the in vivo differentiation ability of Muse cells, cell samples of human fibroblasts and BMSCs were subjected to LTT for 8 h and 16 h, respectively, obtaining MEC. MEC were transplanted into immunodeficient mice with different injuries, and they were found to integrate into the damaged tissue, differentiating and expressing tissue-specific markers. This demonstrates that Muse cells are also capable of differentiating into the three germ layers in vivo [11]. Many other studies have reported spontaneous differentiation in vivo into different cell types, such as cardiomyocytes [30], neuronal and glial cells [39,40,41,42,43], hepatocytes, sinusoid endothelial cells, cholangiocytes and Kupffer cells [17,44,45], vascular endothelial cells [46,47] and glomerular cells [48].

To prove their self-renewal ability, one study transferred MEC to suspension culture and spontaneously formed M-clusters. After that, M-clusters were put on adherent culture, reinitiating cell proliferation. When cultures had expanded to 5–10 × 10^4^ cells, they underwent LTT to obtain MEC and then, they were transferred again to the suspension culture. Approximately 12% of the cells derived from the fibroblasts samples and 8% of cells derived from BMSCs formed M-clusters. The LTT cycle, followed by suspension culture and then adherent culture was repeated up to five times, and each generation had a similar M-cluster formation rate, proving that Muse cells are able to self-renew [11].

Due to their self-renewal ability and triploblastic differentiation capacity, Muse cells have been described as pluripotent cells [11].

### 3.2. Non-Tumorigenicity

A major problem with pluripotent stem cells such as ES cells or induced pluripotent stem cells (iPSCs) is that they have a high multiplication rate and form teratomas in vivo [49].

Although human Muse cells have been shown to be pluripotent, their proliferation rate is not very high; thus, they do not form tumors in vivo. Immunodeficient mice were injected with human M-clusters or MECs, and the results of the study showed that they did not develop teratomas after 6 months [11].

Furthermore, another study showed that genes related to the cell cycle had lower expression in Muse cells than in ES cells and iPSCs. Additionally, both Muse cells and M-clusters had low telomerase activity, indicating that Muse cells may have limited replication potential [23].

### 3.3. Elevated Stress Tolerance and DNA Repair Ability

Muse cells received their name because of their tolerance to stress, as they can survive 16 h LTT without nutrients [11]. Recently, another study has proved their tolerance to physical and chemical genotoxic stress by applying hydrogen peroxide (H_2_O_2_) and UV radiation. Apoptosis was not incremented in Muse cells and additionally, it was observed that UV radiation produced a decrease in apoptosis 48 h after the exposure [26].

Regarding the mechanisms that allow Muse cells to have that tolerance, it has been discovered that they secrete 14-3-3 proteins. These factors play an important role in stress tolerance because they contribute to the regulation of cell cycle and cell response to DNA damage. It is suggested that Muse cells may secrete pro-survival factors acting in an autocrine or paracrine way, as their secretome includes many 14-3-3 isoforms related to anti-apoptotic activity [25]. Moreover, it has been described that the stress-activated protein kinase/Jun amino terminal kinase signaling pathway is also related to Muse cells’ stress tolerance [50].

DNA damage response (DDR) is the process carried out by cells to repair DNA damage and coordinate it with cell cycle progression [51]. After DNA injury occurs, single-strand breaks (SSB) and double-strand breaks (DSB) can be produced. Base pair excision repair (BER) and nucleotide excision repair (NER) are two systems whose function is repairing SSB, while non-end joining recombination (NHEJ) and homologous recombination (HR) repair DSB [52]. Moreover, when DNA is damaged, ataxia-telangiectasia mutated kinase (ATM) is activated and coordinates the repairing response. If DDR works correctly and DNA is repaired, ATM becomes inactive again [51].

In a study conducted by Alessio et al. [26], it was found that ATM activation increased after inducing damage (using H_2_O_2_ and UV radiation) in the DNA, and such activation decreased to basal level 48 h later in Muse cells. On the other hand, in non-muse cells, ATM activation did not occur correctly and, in MSCs, ATM activation levels did not return to basal level by 48 h. The investigation showed that the efficiency in repairing SSB by BER and NER was similar between Muse cells, non-Muse cells and MSCs, whereas NHEJ was higher in Muse cells [26].

Furthermore, the expression of γ-H2Ax, which is the active and phosphorylated form of H2Ax was studied [26]. After DNA damage, γ-H2Ax foci indicate regions that are being repaired until the DNA is completely restored. If DNA has not been correctly repaired, γ-H2Ax foci will still be detected in these zones [52]. The analysis revealed several γ-H2Ax foci 1 h and 6 h after DNA damage induction (using H_2_O_2_ and UV radiation) in Muse cells. After 48 h, γ-H2Ax foci were also observed in Muse cells that were in the G1 phase. However, at this point, the percentage of γ-H2Ax foci detection in MSCs and non-Muse cells was higher than in Muse cells [26].

Due to these factors and mechanisms, Muse cells are able to detect and repair DNA damage within a short period of time and, therefore, have high stress tolerance [26].

### 3.4. Preferential Homing to Damaged Locations

Sphingosine is a component of the cell membrane that synthesizes the sphingolipid sphingosine-1-phosphate (SP1) when a variety of stimuli occurs, such as inflammation or tissue damage [53,54]. Although Muse cells express the five subtypes of the SP1 receptor (S1PR), sphingosine monophosphate receptor 2 (S1PR2) is the most represented [30].

The S1P-S1PR2 axis is the mechanism that allows specific homing of Muse cells into the damaged area, regardless of the tissue or organ damaged. To confirm this hypothesis, the S1PR2 antagonist JTE-013 was administered at the same time as the Muse cells. The specific homing of these cells did not happen, verifying this selective homing system. It was also proved that the S1PR2 agonist SID46371153 improves Muse cells migration in vitro [30].

Moreover, the preferential homing of Muse cells has been supported by other studies. In a rabbit model of AMI, 14% of Muse cells administered intravenously engrafted into the infarcted area whereas the same procedure with MSCs led to only a few cells engrafted [30]. Additionally, Muse cells have been reported to be better at selective homing than MSCs or non-Muse MSCs in rat models of middle cerebral artery occlusion ischemia [39] and perinatal hypoxic ischemic encephalopathy [43], as well as in mouse models of lacunar stroke [55], doxorubicin-induced nephropathy [48], amyotrophic lateral sclerosis [42], Shiga toxin-producing *Escherichia coli*-associated encephalopathy [41] and epidermolysis bullosa [56].

Finally, it has been demonstrated that after an AMI, Muse cells are mobilized into peripheral blood following the increase in plasma S1P level. A study was performed including 79 human patients with AMI, 44 patients suffering from coronary artery disease (CAD) and 64 healthy patients. The results showed no significant differences in the number of circulating Muse cells between the groups on the day of admission (day 0). There was a peak on day 1 and, finally, the number of Muse cells decreased by day 14 and 21. The Muse cell number was significantly higher 14 days after the AMI compared with that of the CAD and control group. Furthermore, the study showed that an increase in the level of SP1 proceeded elevation in Muse cells number and described a positive correlation between them, suggesting that SP1 can mobilize Muse cells into the peripheral blood [57].

### 3.5. Immunosupression, Vascularization, Anti-Fibrosis, Anti-Apoptosis and Anti-Inflammation

It has been demonstrated that soluble mediators secreted by human Muse cells are responsible for the immunomodulatory capacity of these cells. Transforming growth factor-β1 (TGF-β1) has been identified as an important mediator of Muse cells immunomodulatory actions in macrophages and T lymphocytes. It has also been reported that an increase in TGF-β1 level in M-clusters was associated with low cell proliferation, suggesting that the TGF-β1 autocrine/paracrine loop may be related to the low proliferative rate in Muse cells [58].

Additionally, Muse cells have immunosuppressive properties, as they activate regulatory T cells, suppress dendritic cells differentiation and express human leukocyte antigen G (HLA-G) [30]. HLA-G was detected in immune-privileged organs and is related to reduction in inflammation and immune responses [59]; as well as present in placental mammals to avoid fetus and placenta rejection [60]. It has also been reported to support tolerance in heart transplantation [61]. For this reason, although HLA-G expression is lower after engrafting into the damaged area, it is suggested that HLA protects Muse cells from immunologic response [29]. Muse cells also express the indoleamine 2,3-dioxygenase (IDO), an enzyme that is involved in immunosuppression [60].

Moreover, other factors secreted by Muse cells have a key role in anti-apoptosis, neovascularization and stimulation of cardiomyocyte progenitors. These components are hepatocyte growth factor (HGF) and vascular endothelial growth factor (VEGF) [30,62], which are better expressed by Muse cells than by MSCs [17,30]. It has been reported that HGF and VEGF protect liver sinusoidal endothelial cells and reduce necrosis in mouse and swine liver damage models treated with Muse cells [32,45]. Additionally, the suppression of these factors by siRNA in Muse cells led to deterioration of protection effects [17].

Nevertheless, these are not the only pathways for Muse cells to promote neovascularization, as they can differentiate in vivo into vascular cells in the damaged area in animal models of chronic kidney disease, aortic aneurysm, AMI and liver damage, as well as in diabetic skin ulcers in a diabetic immunodeficient mouse model [17,30,46,47,48].

Muse cells also produce matrixxmetalloproteases-1 (MMP1), MMP2 and MMP9 [45], which are responsible for antifibrosis/fibrinolytic effects because these enzymes degrade the extracellular matrix [63,64]. Different studies have demonstrated reduction of fibrosis using Muse cells in animal models of liver fibrosis, chronic kidney disease and AMI [30,45,48].

Regarding their anti-inflammatory, anti-apoptotic and tissue protection functions, Muse cells produce other factors and cytokines, including epidermal growth factor (EGF), keratinocyte growth factor (KGF), platelet-derived growth factor subunit A (PDGFA), platelet-derived growth factor subunit B (PDGFB), fibroblast growth factor 2 (FGF2), fibroblast growth factor 6 (FGF6), Angiopoietin 1 (ANGPT1), nerve growth factor (NGF), brain-derived neurotrophic factor (BDNF), glial cell line-derived neurotrophic factor (GDNF), insulin-like growth factor 1 (IGF-1), interleukin-1 receptor antagonist (IL1RA), stem cell factor (SCF), stromal cell-derived factor 1 (SDF-1), prostaglandin E2 (PGE2) and interleukin-4 (IL-4), IL-10, IL-13 [29]. Furthermore, Muse cells increased the production of anti-inflammatory cytokines (IL-10, PGE2, TGF-β, tumor necrosis factor-α-stimulated gene/protein-6 (TSG-6)) and growth factors (IGF-1, VEGF-A, VEGF-B, VEGF-C, VEGF-D, EGF, TGF-α), as well as decreased the expression of inflammatory cytokines (IL-1B, IL-6, IL-17A, IL-33, tumor necrosis factor (TNF)) in different animal models [32,47,48,65,66,67].

Recently, it was demonstrated that Muse cells regulated the phenotype of microglia in vitro in an inflammatory microenvironment, increasing the number of M2 microglia and producing a decrease in the proportion of M1 microglia, in a more effective way in comparison with BMSCs. Additionally, the antineuroinflammatory effect of Muse cells has been reported, as they suppress the TLR4/MyD88/NF-κB and p38 MAPK signaling pathways in microglia [68].

Regarding Muse cell effect in cocultures, when these cells were cocultured with human T cells, there was an increase in expression of IL-10 and CD25 in T cells. It has also been demonstrated that when Muse cells were cocultured with monocytes or monocyte-dendritic cell progenitors, the differentiation of monocytes was suppressed [30]. Another experiment showed that coculture of Muse cells and macrophages reduced the expression of TNF-α. What is more, it has been reported that Muse cells reduced production of IL-6 and interferon-γ (IFN-γ) and increased TGF-β and IL-10 when they were cocultured with intestinal epithelial crypt cells damaged by TNF-α, showing protective effects on intestinal barrier [31].

Due to all these immunomodulatory properties, as well as their other unique characteristics, Muse cells may be useful in the treatment of inflammatory diseases [29].

## 4. Comparison between Muse Cells and MSCs

First, there are differences between marker expression of Muse cells and MSCs. In general, MSCs have been described as CD105+, CD73+ and CD90+ [15]; whereas Muse cells have been reported to be SSEA-3+, CD105+, CD45+, CD29+ and CD90+, depending on the tissue from which they were obtained [11,23,28]. Additionally, single Muse cells were reported to form M-clusters similar to embryoid bodies formed by ES cells in suspension culture, whereas individual non-Muse cells were not able to survive in suspension culture [11].

Muse cells were reported to perform both symmetric and asymmetric cell division, while non-Muse cells were generated by the second type [11]. As non-Muse cells account for approximately 99% of MSCs, it has been suggested that the actions of non-Muse cells represent the majority of those observed in MSCs and, thus, the results obtained using Muse cells would be better [15].

MSCs are somatic stem cells that can be isolated from different tissues in the adult organism, such as bone marrow or adipose tissue [15]. Although they have been reported to differentiate into cells of the three germ layers by cytokine induction or gene introduction [69,70,71,72], the differentiation rate was not high, and some authors have suggested that this ability is only provided by a small subpopulation among MSCs [73]. These cells have been reported to be Muse cells, which are endogenous pluripotent stem cells that can also be harvested from a variety of tissues [11]. The triploblastic differentiation ability of Muse cells has been demonstrated both in vitro [23,36,37,38] and in vivo [11,17,30,39,40,41,42,43,44,45,46,47,48]. In contrast with MSCs, the rate of induced differentiation of Muse cells is very high, accounting for 80–95% [29]. Since differentiation potential is directly related to the efficacy of the treatment, MSCs have limited tissue regeneration capacit, and Muse cells appear to be a better option due to their spontaneous differentiation ability [15].

Regarding the migration and integration into the damaged area, local or intravenous injection of MSCs led to a few cells homed into the injured zone and differentiated into tissue-specific cells. Several studies have reported that Muse cells were better at selective homing than MSCs or non-Muse MSCs in different animal models [30,39,41,42,43,48,55,56]. The S1P-S1PR2 axis has been described as the mechanism that allows specific integration of Muse cells into the injured zone [30].

Furthermore, Muse cells have been reported to be stress tolerant, providing an important advantage over MSCs. Muse cells survived 16 h LTT without nutrients [11], and they showed greater resistance to chemical (H_2_O_2_) and physical (UV radiation) genotoxic stress. While neither of these two treatments increased apoptosis in Muse cells, MSCs incremented apoptosis after treatment with H_2_O_2_. Additionally, although the DNA repairing systems BER and NER showed similar efficacy between Muse cells and MSCs, NHEJ was greater in the first ones [26]. Since damaged tissue is considered a hostile environment, especially in the acute or subacute phases of a disease, Muse cells could increase survival rate due to their stress tolerance [15].

Finally, a limited anti-fibrotic effect of MSCs and non-Muse MSCs compared to Muse cells was postuled; because this process is necessary, especially after the acute phase, and they do not remain in the tissue long enough [30,32,48,65].

## 5. Use of Muse Cells in the Treatment of Various Diseases

### 5.1. Acute Myocardial Infarction

It has been postulated that an increase in the number of peripheral circulating Muse cells could be a prognostic factor in AMI, as they are correlated with avoidance of heart failure and functional recovery of the heart muscle [74]. After concluding that, the net hypothesis could be if administration of Muse cells after AMI results in a higher recovery.

In this regard, it has been demonstrated that intravenously administration of a single dose of autograft Muse cells (3 × 10^5^ cells/2 mL of saline) improves left ventricular (LV) function, reduces the infarct size and mitigates left ventricle remodeling in a rabbit model of AMI. The histologic findings also revealed a high engraftment rate of the Muse cells into the infarcted site and border area (14.5% ± 4.0% of the injected Muse cells). Additionally, it was proven that the specific S1PR2 antagonist JTE-013 blocked the engraftment of Muse cells, suggesting that this process is mediated through the S1P-S1PR2 axis [30], as described above in the Muse cell properties.

Moreover, autograft, allograft and human GFP (green fluorescent protein)-labeled Muse cells engrafted to the infarct site and border areas expressed some cardiac markers such as ANP, troponin I and α-actine, as well as CD31 and α-smooth muscle actin (a vascular endothelial marker and a vascular smooth muscle marker, respectively). These findings suggest that after Muse cells engrafted in the tissue, they transdifferentiated into cardiomyocytes and vessels spontaneously. What is more, it was demonstrated that Muse cells differentiated into cardiomyocytes with physiologic activity, as GCaMP3 (GFP-based Ca calmodulin probe)-Muse cell-derived cardiomyocytes show lower GCaMP3 fluorescence during diastole and higher fluorescence during systole, synchronous to an electrocardiogram [30].

Another investigation revealed that Muse cell therapy was effective and safe in a mini-pig model of AMI and, therefore, may be a potential option for AMI treatment. The animals in the Muse group received 1 × 10^7^ human Muse cells/2 mL while 2 mL of saline were administered to the control group. After 12 weeks, the ejection fraction and fractional shortening were significantly better in the Muse group than in the control group. Additionally, LV end-diastolic dimension and LV end-systolic dimension were significantly reduced in the Muse group in comparison with the control group. The infarct area was also smaller in the Muse group (10.5% ± 3.3% of the LV) compared with the control group (21.0% ± 2.0% of the LV). In the Muse group, GFP-labeled Muse cells were mostly identified in the infarct border area and expressed cardiac troponin I. These GFP-labeled Muse cells from the infarct border area were CD31+ (a marker of vascular endothelial cells). Finally, blood cell count and biochemical analysis were performed and did not show any abnormality in either group [22].

Recently, a new study conducted in an ovine model of AMI has demonstrated that an intramyocardial injection of allogenic Muse cells improves neovascularization and survival rates. Animals were divided into two groups and 1 × 10^7^ allogenic Muse cells were administered to the Muse group while the control group received only phosphate buffer saline (PBS). Previously to the intramyocardial injection, Muse cells were labeled with PKH26 Red Fluorescent Dye. These labeled-Muse cells were detected in the infarct border area 7 days after the AMI, as well as markers of cardiac differentiation (desmin, sarcomeric actin and troponin T) in the same zone in the Muse group. The marker lectin was also localized in the same area as PKH26+ cells, suggesting formation of new vessels. In fact, the density of arterioles and capillaries in the infarct border area was significantly greater in the Muse group than in the control group. Additionally, the study reported a higher gene expression of VEGF in the Muse group in comparison to the control group [35].

Regarding the administration of Muse cells to humans, the first-in-human clinical trial with 3 patients with ST-elevation myocardial infarction (STEMI) was conducted. The allogenic Muse cell-based product CL2020, composed of 1.5 × 10^7^ cells/15 mL of frozen preparation, was administered to the three STEMI patients. Any adverse side effects related to the administration were detected for up to 12 weeks. No arrhythmias, thrombotic disorders, shock or infections were detected in that period. Furthermore, left ventricular ejection fraction measured by echocardiography significantly increased from 40.7% ± 1.5% to 52.0% ± 2.6% after 12 weeks. The Wall motion score index was significantly decreased after 8 and 12 weeks from the CL2020 administration, which could indicate that regional LV systolic function was significantly increased. The authors concluded that the CL2020 administration to STEMI patients was safe and improved LV function [75].

### 5.2. Stroke

Stroke is a cerebrovascular disease considered the second cause of death in the world. It is produced by arterial occlusion or rupture of blood vessels. Although 80% of the cases are ischemic stroke, it can also be hemorrhagic. Due to its relevance, different studies have reported clinical applications of Muse cells in stroke [21].

One study was performed in immunodeficient mice in which permanent middle cerebral artery occlusion was induced. These animals were divided into four groups: Muse, non-Muse, BMSC and control group. BMSCs were obtained from human BM, as well as Muse and non-Muse cells, which were isolated within this BMSCs population. The number of cells for each transplantation was 2.5 × 10^4^ cells in Muse, non-Muse and BMSC groups, while animals in the control group only received 10 μL of PBS. The cells were transplanted into the ipsilateral striatum 7 days after the induction of the disease. Then, 21 days after the administration, animals in the non-Muse group started recovering motor functions. However, functional recovery in this group was not promoted after day 42. These results corroborated the histologic findings, because non-Muse cells were not found in the injured area after day 42. Therefore, non-Muse cells were suggested to be useful in the acute phase. On the other hand, Muse cell effects were observed after day 35, when mice started the recovery of motor function. Regarding the histological findings, the number of remaining cells in the damaged area was significantly increased in the Muse group than in the non-Muse and BMSC groups. Finally, a study showed that most Muse cells homed into the infarcted zone and expressed Tuj-1 and NeunN (neuron-specific markers) at day 42, indicating that Muse cells may prefer neuronal-lineage differentiation. The authors suggest that non-Muse cells have paracrine effects and Muse cells improve functional recovery, as well as neuronal tissue regeneration [76].

Other researchers have also reported clinical benefits of the application of Muse cells to a rat model of transient middle cerebral artery occlusion. In this case, Muse cells were obtained from human dermal fibroblasts. Animals were divided into three groups: Muse, non-Muse and control groups. Each animal in the Muse and non-Muse groups received three injections in 3 different locations, and each injection had a concentration of 1 × 10^4^ cells/2 μL of PBS while only PBS was administered to the animals from the control group. The study confirmed that Muse cells contributed to functional recovery, measured with the rotarod test and modified neurologic severity score by day 70 and 84, whereas this functional recovery was not detected in the other two groups. Moreover, Muse cells were found to survive in the surrounding area of the infarct for up to 84 days, in contrast with non-Muse cells. Muse cells were positive to NeuN (64.6%  ±  0.6% of GFP+ cells), MAP-2 (32.4%  ±  2.4%), interneuron marker calbindin (27.5%  ±  2.5%) and GST-π (25.0%  ±  0.8%, an oligodendrocyte marker). The retrograde tracing experiment revealed that Muse cells integrated into the sensory-motor cortex and then they extended neurites into the pyramidal tract, at least to C1 level. Muse cells that integrated into the sensory cortex participated in the sensory neural circuit. The author concluded that Muse cell benefits were related to the reconstruction of the damaged neuronal circuit [39].

Another investigation was performed in an immunodeficient mouse model of subacute lacunar stroke. Muse cells were obtained from human BMSCs and GFP-labeled. Three groups were formed: Muse, MSC and control groups. Muse cells were transplanted into the peri-lesion area in a concentration of 1 × 10^5^ cells/3 µL PBS. The MSC group received the same number of cells, but they were serum/seno-free MSC. The control group only received PBS. Approximately 28% of Muse cells homed into the brain, most of them specially into the damaged area, whereas only a few MSCs were detected at 8 weeks post-treatment. These Muse cells engrafted near the damaged zone expressed NeuN (62.2 ± 2.4% of GFP positive cells) and MAP2- (30.6 ± 3.1%). They also differentiated into GST-π-positive cells (12.1 ± 1.1%), suggesting that Muse cells not only differentiated into neuronal cells but also into oligodendrocytes. Moreover, Muse cells contributed to the reconstruction of the pyramidal tract. Regarding the clinical outcomes, Muse cells led to a significant behavior improvement in comparison with the other two groups after 6 and 8 weeks of transplantation. Additionally, no tumors were found up to at least 6 months, indicating that Muse cells may be a safe option for subacute lacunar stroke treatment [55].

Finally, another research study was conducted on an immunodeficient mouse model of lacunar stroke studying the administration of CL2020 (a product of Muse cells). All the animals were injected into the cervical vein, either at the subacute phase (around 9 days) of the lacunar stroke or in the chronic phase (around day 30). Animals injected in the subacute phase were divided into 3 groups: high-dose (5 × 10^4^ Muse cells), low-dose (5 × 10^3^ Muse cells) and control group. Animals that received the treatment in the chronic phase were divided into high-dose (5 × 10^4^ Muse cells), medium-dose (1 × 10^4^ Muse cells) and control group. Muse cells homed specifically to the peri-infarcted area for 10 (subacute) or 16 (chronic) weeks, expressing NeuN and MAP-2 markers. The high-dose group had significantly better functional recovery than the control group (both subacute and chronic phase animals) up to 22 weeks. No tumorigenesis was observed during the study [77].

### 5.3. Amyotrophic Lateral Sclerosis

Amyotrophic lateral sclerosis (ALS) is a neurogenerative disease in which there is a progressive loss of motor neurons, and benefits of the approved treatments for this illness are still limited [42].

To prove the efficacy of Muse cell therapy in this disease, transgenic mice with the G93A human SOD1 mutation were used and divided into three groups (Muse, MSCs and control group). Animals were injected into the tail vein with a total of 10 doses from 56 days of age until 119. In the Muse group, the composition of each administration was 5.0 × 10^4^ Muse cells/250 μL Hank’s balanced salt solution (HBSS), while the composition was the same but using MSCs in the MSC group. Both MSCs and Muse cells were isolated from human BM. GFP-labeled cells were also produced [42].

In the rotarod test, the hanging-wire score and muscle strength of the lower limbs significantly improved in the Muse group in comparison with the control group. GFP-Muse cells were detected at the spinal pia-mater and at the ventral horn, whereas a smaller number of GFP-MSCs were only found in pia-mater after 22 weeks. Additionally, 85.7% of GFP-Muse cells co-expressed GFAP (an astrocytic marker) at the pia-mater and ventral horn. These findings suggested that most of the Muse cells differentiated into astrocyte-lineage [42].

A significantly higher number of motor neurons was observed in the Muse group compared with the control group, while there was no significant difference between the MSC and control group. Muse cells significantly improved values for myofiber size compared with the MSC and control group, in which severe neurogenic myofiber atrophy was detected [42].

The study revealed clinical benefits when using Muse cells in the ALS mouse model, as these cells homed into the spinal cord, provided astroglia, helped motor neuron survival and avoided myofiber atrophy [42].

### 5.4. Aortic Aneurysm

The aorta is a structure with different layers composed of several types of cells, such as fibroblasts, endothelial cells (ECs) and vascular smooth muscle cells (VSMCs), as well as an extracellular matrix (ECM). This matrix includes elastic and collagen fibers, among other components. The aortic aneurysm (AA) is a progressive enlargement of the aorta that becomes more dangerous as the diameter increases [78].

Recently, a study has discovered that human BMSC-Muse cells injected intravenously to immunodeficient mice with induced abdominal AA in the acute phase resulted in an improvement of the disease. Muse cells can migrate into the AA and differentiate into ECs and VSMCs, as well as preserve elastic fibers, producing a decrease in the dilatation [47].

The in vitro experiment confirmed that Muse cells can differentiate into both VSMCs and ECs. Although CD34+ progenitor cells include vascular progenitor cells, differentiation into ECs and VSMCs is only possible under cytokine induction, and expression of VSMC differentiation-related markers was lower in CD34+ progenitor cells than in Muse cells [47].

The dilatation ratio in the group treated with multiple injections of Muse cells (M) was lower than in the groups treated with non-Muse cells (N), MSCs (MSC) and vehicle group (V) 3 weeks after the injection. At that point, there was no significant difference between the M group and group treated with single injection of Muse cells (M’). Furthermore, 8 weeks after the injection, there was still no difference between the M and M’ group, but the dilatation ratio in both of them was significantly smaller than the V group. Moreover, dilatation in the N, MSC and V group did not show significant differences between them at 8 weeks. Each injection of Muse cells contained 20,000 cells and was dissolved in 0.2 mL of PBS [47].

The histologic findings revealed that 3 weeks after the injection, the aortic medial elastin content was significantly higher than the results in N, MSC and V groups. The measures in M and M’ groups showed no significant difference while the ratio in N, MSC and V was also not significantly different [47].

Additionally, the researchers demonstrated that on day 3, Muse cells were detected only at the outermost layer of the vasculature. Some Muse cells were located around a vasculature structure similar to the vasa vasorum at the tunica externa. Moreover, Muse cells not only homed in the tunica externa, but then migrated towards the tunica media and luminal layers by day 5 [47].

Due to all these results, Muse cells can substantially attenuate the abdominal AA dilatation in the acute phase. Further studies are needed to clarify their utility in chronic aneurysms, other animal models and their application in thoracic AA [47].

### 5.5. Type 1 Diabetes

It has been reported that patients with type 1 diabetes have dysregulation in the immune system that contributes to disease progression. Due to the immunomodulatory properties of Muse cells, it has been suggested that if muse cells could suppress the pro-inflammatory surge, more insulin-producing cells would survive [58].

Diabetic NOD mice were used as a model of spontaneous autoimmune diabetes and divided into a Muse group and control group. An intraperitoneal injection of 1 × 10^6^ Muse cells harvested from the adipose tissue was administered to the animals in the Muse group, while only sterile PBS was injected in the control group [58].

The results revealed that blood glucose levels varied between 202 and 500 mg/dL in the Muse group in a 7-week period after cell administration, whereas glucose levels increased and remained above 500 mg/dL in the control group after a week. Additionally, mice treated with Muse cells kept body weight stable for 7 weeks, while mice from the control group lost considerable weight in the 2 weeks following PBS administration [58].

Muse cells may be useful in the treatment of spontaneous diabetes by controlling the autoimmune process. The authors affirmed that analyzing how Muse cells improve glycaemia control would be interesting. They also suggested that Muse cells differentiation into insulin-producing cells might be possible in the animal model [58].

### 5.6. Diabetic Skin Ulcers

In addition to the administration of Muse cells to treat diabetes, another study has demonstrated that a subcutaneous injection of Muse cells can be a good method to improve healing of diabetic skin ulcers. Muse cells were isolated as SSEA-3+ cells from human AMSCs and administered to diabetic immunodeficient mice. The results confirmed that the group treated with Muse-rich cells showed significantly faster wound healing compared to the ones treated with Muse-poor cells. Additionally, the study revealed that 14 days after the injection of Muse-rich cells, the wounds had fully healed, whereas in the group treated with Muse-poor cells the wound size was still 30.3% ± 6.7% [46].

Furthermore, the histological assays revealed that Muse-rich cells had integrated into the dermis tissue either as cells of the vascular endothelium, or as cells of the upper and lower dermis [46].

### 5.7. Chondral Injuries

Recently, it has been demonstrated in immunodeficient rats that human BMSC-Muse cells administered intra-articularly (5 × 10^4^ cells/50 µL of PBS) improve the repair of chondral tissue after the generation of a defect, compared to the injection of MSCs or non-Muse cells [79].

Macroscopically, in the Muse group, the chondral defect made in the knee of the rat was filled with white tissue and the surface seemed smooth and homogeneous, whereas the histologic findings in the cartilage were not satisfactory. However, both macroscopic and histological results were better in the Muse group than in the other groups 12 weeks after the treatment [79].

Nevertheless, there is a need for future research into Muse cells and their chondrogenic potential [79].

### 5.8. Atopic Dermatitis

Atopic dermatitis (AD) is an inflammatory skin disease in which dysregulation of the immune system and epidermal barrier dysfunction play an important role [80]. This disease produces chronic pruritus (CP) in 100% of the patients [81].

To assess the utility of Muse cell administration to AD patients, 2 × 10^5^ human Muse cells were dissolved in 20 µL of PBS and injected subcutaneously to mice models of AD in the Muse cell group, while only PBS was administered to the control group. After the Muse cell administration, redness of the skin and itching and scratching behaviors were relieved, as well as the wounds caused by scratching, which healed better in comparison with the PBS group. The number of scratches in the Muse cell group decreased to baseline by day 9, whereas this number in the control group was still significantly increased by day 10 [65].

Moreover, the migration of Muse cells after subcutaneous injection was studied. The immunofluorescence analysis revealed that Muse cells progressively migrated from the injection point to the damaged skin area [65].

Additionally, the authors studied microglial activation in both groups and found that it was significantly decreased in the Muse group compared with the control group. Furthermore, the expression level of different inflammatory factors (IL-17α, IL-6, and IL-33) was measured in the back skin and spinal dorsal horn. The results showed that the level of these factors significantly decreased after the administration of Muse cells [65].

As described above, the treatment with Muse cells reduced scratching behavior and improved cutaneous healing. To corroborate that skin regeneration was produced directly by Muse cells and not by reduced scratching, a model of skin wound injury was used. Muse cells administered subcutaneously improved healing of the cutaneous wound in comparison to the control group [65].

Due to all these findings, the authors concluded that administration of Muse cells can protect the skin in chronic itching illnesses [65].

### 5.9. Adriamycin Nephropathy

Adriamycin (doxorubicin hydrochloride) nephropathy is an established mice model of chronic proteinuric kidney disease with similarities to focal segmental glomerulosclerosis (FSGS) in humans. The investigators used both immunodeficient (SCID) and non-immunodeficient (BALB/c) mice in the study to avoid results affected by the immune response of the animal. Muse cells and non-muse cells were isolated from human BMSCs, and GFP-labeled cells were produced [48].

SCID and BALB/c mice were both divided into three groups: Muse, non-Muse and control group. Muse and non-Muse groups were administered a single intravenous injection of 2 × 10^4^ Muse cells or non-Muse cells, respectively, suspended in sterile saline, while the control group only received sterile saline. Additionally, 2 × 10^4^ GFP-labeled cells were infused for immunohistochemistry [48].

In both SCID and BALB/c models, Muse cells showed specific homing into injured glomeruli, while non-Muse cells did not present that preferential migration and were observed in the spleen and lungs. Muse cells also spontaneously differentiated into cells that expressed CD 31 (marker of endothelial cells, 41%), podocin (marker of podocytes, 31%) and megsin (marker of mesangial cells, 13%) [48].

Although Muse cells disappeared 5 weeks after the administration, the decrease in glomerular sclerosis and interstitial fibrosis was maintained in the Muse BALB/c group in comparison with the non-Muse and the control group by week 7 [48].

Finally, functional restoration of the creatinine clearance, urine protein-to-creatinine ratio and plasma creatinine was detected in the Muse BALB/c group compared with the control group 5 weeks after administration. However, no significant differences were found between the three groups at 7 weeks [48].

### 5.10. Liver Fibrosis

Cirrhosis and other chronic liver diseases are characterized by fibrosis and a reduction in the number of functional cells. Liver transplantation is considered the most effective treatment, but stem cell therapy is an alternative option in the treatment of these severe liver pathologies [45].

The liver fibrosis model was developed using SCID mice. Animals were divided into three groups: Muse, non-Muse and control group. The Muse group received 3 intravenous injections containing 5 × 10^4^ Muse cells per injection. The same quantity of non-Muse cells was administered to the non-Muse group while only PBS was injected in the control group. Muse cells and non-Muse cells were obtained from human BMSCs [45].

Muse cells spontaneously differentiated into cells positive to DKL, CK18, CK19 and α-fetoprotein (markers of hepatoblasts or hepatocytes) on gelatin culture. When administered to the animal model, Muse cells specifically homed into the liver and differentiated into cells that expressed different markers, such as human anti-trypsin, human albumin, HepPAr-1, human CYP1A2 (a detoxification enzyme) and human Glc-6-Pase. This marker expression, as well as improvement in total bilirubin and serum albumin, suggested that Muse cells spontaneously differentiated into human hepatocytes in vivo, enhancing liver function [45].

Regarding the histological findings, Muse cells were reported to reduce fibrotic areas, either by fibrolysis or attenuation of fibrosis, compared with non-Muse cells. Muse cells remained in the liver up to 8 weeks, whereas non-Muse cells did not. Additionally, greater levels of MMP9 expression were detected in the Muse group than in the non-muse group, suggesting a role of this protein in the reduction of fibrosis [45].

Due to all these promising results, Muse cells may be useful for the treatment of chronic liver diseases [45].

### 5.11. Ischemia-Reperfusion Lung Injury

A frequent cause of primary graft dysfunction after lung transplantation is ischemia-reperfusion (IR) lung injury, which accounts for 25% of the deaths [82].

To study the effect of Muse cell treatment on patients with this syndrome, a lung IR injury rat model was used. Muse cells were obtained from human BMSCs. The animals were divided into three groups: the first one received a single dose of 1.5 × 10^5^ human Muse cells/200 μL PBS through the left pulmonary artery (Muse group), the same concentration of human MSCs were administered through the same injection route to the second group (MSC group) and only 200 μL of PBS were injected to the control group [83].

The analysis of arterial oxygen partial pressure, fractional inspired oxygen ratio, alveolar-arterial oxygen gradient and lung compliance showed better results in the Muse group than in the MSC group on day 3 and 5 [83].

Moreover, alveolar edema, intra-alveolar hemorrhage, capillary congestion and neutrophil infiltration in the Muse group were lower than in the MSC and control group by day 3. The number of human Golgi+ cells was greater in the Muse group than in either the MSC or control group on day 3 and 5, showing that Muse cells remained in the injured tissue in a more efficient way in comparison with MSCs. Additionally, the number of TUNEL-positive cells was lower in the Muse group in comparison with that in the MSC or control group on day 3, and the number of TUNEL-positive cells decreased even more by day 5, demonstrating the greater apoptosis-suppressive capacity of Muse cells compared with MSCs [83].

The number of alveolar type II cells was higher in the Muse group than in the MSC and control group by day 3. Five days following injection, no significant differences between the Muse and MSCs groups were reported. However, the number of alveolar-type II cells was significantly higher in the Muse group than in the control group [83].

Finally, the authors suggested that Muse cells improved pulmonary function and histologic damage in the acute phase after IR lung injury, possibly due to suppression of apoptosis, promotion of type II alveolar epithelial cell proliferation and protection of lung tissue [83].

## 6. Future Directions

The biological properties of Muse cells make them ideal candidates for application in regenerative medicine. For this reason, it is particularly important to develop animal models to test their efficacy. The next step should be to increase our knowledge of Muse cells and characterize them in more animal species. Additionally, it would also be interesting to characterize these cells in more tissues in each species, following the example of humans and mice. Furthermore, future studies should evaluate the effectiveness of Muse cells in other diseases, since they have only been tested in a small number of pathologies. Equally interesting would be to carry out studies using allogeneic Muse cells for a closer approximation to their clinical use.

## 7. Conclusions

Muse cells have been isolated and characterized in humans and other animal species. These cells are non-tumorigenic endogenous pluripotent stem cells that present elevated stress tolerance and high capacity to repair DNA damage. Muse cells can spontaneously differentiate into cells of the three germ layers, have self-renewal properties and are able to specifically home into the damaged area. Due to their unique biological properties, the effectiveness of Muse cells has been studied and confirmed in a variety of diseases, such as AMI, stroke, aortic aneurysm and liver fibrosis.

## Data Availability

Not applicable.

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
