# Peer review of "Multilineage-Differentiating Stress-Enduring Cells (Muse Cells): The Future of Human and Veterinary Regenerative Medicine"

_biomedicines, 2023, doi:10.3390/biomedicines11020636_

Round 1
Reviewer 1 Report
Dear authors,
The work is really interesting, well-written. In the revision about Muse cells ia actual, and the topic is well described an explained.
The only suggestion is discussed about the isolation of Muse cells using sorting CD105 and SSEA-3 positive cells.
Author Response
RESPONSE TO REVIEWER 1
Dear authors,
The work is really interesting, well-written. In the revision about Muse cells is actual, and the topic is well described an explained.
The authors would like to thank the reviewer for the positive comments about the work as well as for the advice that helped to improve the review.
The only suggestion is discussed about the isolation of Muse cells using sorting CD105 and SSEA-3 positive cells.
A paragraph explaining when is used the sorting by SSEA-3/CD105 double positive cells and when is useful the isolation as SSEA-3+ cells has been included (lines 94-99):
“Finally, returning to the isolation of Muse cells, it has been reported that Muse cells can be isolated directly from tissue samples as SSEA-3/CD105 double-positive cells by fluorescence-activated cell sorting (FACS) [11,23]. However, isolation as SSEA-3+ cells has been reported to be successful when the basal population is composed of cultured mesenchymal cells (e.g., BMSCs) since nearly all cells will be positive for mesenchymal markers such as CD105 [15].”
Reviewer 2 Report
The Review is devoted to the research of multilineage-differentiating stress-enduring cells (Muse cells), which is very much interesting as this type of cells might be characterized as pluripotent, non-tumorigenic, tolerant to stress, able to repair damaged DNA and because of it is a perspective source for the therapeutic applications.
Authors made intense research, the obtained information is deeply analysed, the manuscript is accurately and logically structured, so the reader would consequentially get the information – at first, an explanation of the studied object is given, then multilineage-differentiating stress-enduring cells biological properties are characterized, the differences between Muse cells and mesenchymal stem cells (MSCs) are presented and finally the contribution of multilineage-differentiating stress-enduring cells to the treatment of some diseases is revealed. Especially comparison between Muse cells and MSCs is quite interesting, because it helps to understand advantages and disadvantages of the Muse cells which might be used for disease treatment.
It is worth noting, that Table 1 is quite informative and illustrative.
The following comments do not diminish the value of the Review:
Probably graphical abstract should be added, as it would help to understand the idea of the Paper at a glance.
Also, it would be brilliant if the figures would be added, as the illustrations would help to understand the information better.
It also would be good if Discussion section would be added, which would include the analysis of the presented information, and it would be quite useful to add the section Future Direction to demonstrate the thoughts about the questions which remained unanswered.
Please check the following through the text – the dots at the ends of the paragraphs’ titles should be removed (lines 3, 61, 64, 94, 110, 132, 146, 156, 171, 188, 199, 204, 232, 244, 283, 313, 371, 414, 415, 474, 541, 567, 604, 622, 634, 646, 672, 696, 721).
Line 94 It would be better to locate the title on the same page with the paragraph content.
Line 158 LTT abbreviature should be deciphered.
Line 194 It would be better to decipher the following abbreviations: ANG and PGF.
Line 233 It would be better to decipher the following abbreviation: ES.
Line 241 The following abbreviation ‘iPS’ should be checked.
Line 269 In the following phrase ‘in MSCs levels did not return to’ it would be better to specify levels of what.
Line 313 It would be better to locate the title of the paragraph on the same page with its content.
Line 348 It would be better to decipher the following abbreviations: ‘KGF, PDGF-A, PDGF-B, FGF-2, FGF-6, ANGPT-1, NGF, BDNF, GDNF, IGF-1, IL1RA’.
Line 349 It would be better to decipher the following abbreviations: ‘SCF, SDF-1’.
Line 350 It would be better to decipher the following abbreviation: ‘PGE2’.
Line 351 It would be better to decipher the following abbreviation: ‘TSG-6’.
Line 353 It would be better to decipher the following abbreviation: ‘TNF’.
Line 455 The following word ‘de’ should be checked.
Line 610 The following fragment ‘1 x 106’ should be corrected.
Line 717 It would be better to decipher the abbreviature ‘MMP9’.
Line 742 It would be better to start the following sentence ‘5 days following injection’ with the word.
Would you please check, the references should be described according the requirements published on the Journal’s website.
Author Response
RESPONSE TO REVIEWER 2
The Review is devoted to the research of multilineage-differentiating stress-enduring cells (Muse cells), which is very much interesting as this type of cells might be characterized as pluripotent, non-tumorigenic, tolerant to stress, able to repair damaged DNA and because of it is a perspective source for the therapeutic applications.
Authors made intense research, the obtained information is deeply analysed, the manuscript is accurately and logically structured, so the reader would consequentially get the information – at first, an explanation of the studied object is given, then multilineage-differentiating stress-enduring cells biological properties are characterized, the differences between Muse cells and mesenchymal stem cells (MSCs) are presented and finally the contribution of multilineage-differentiating stress-enduring cells to the treatment of some diseases is revealed. Especially comparison between Muse cells and MSCs is quite interesting, because it helps to understand advantages and disadvantages of the Muse cells which might be used for disease treatment.
The authors would like to thank the reviewer for all the positive comments about the work as well as for the advice that helped to improve the review.
It is worth noting, that Table 1 is quite informative and illustrative.
The following comments do not diminish the value of the Review:
Probably graphical abstract should be added, as it would help to understand the idea of the Paper at a glance.
Also, it would be brilliant if the figures would be added, as the illustrations would help to understand the information better.
The authors agree with the reviewer that a graphical summary would help to understand the idea of the document at a glance, so it has been added. Regarding the inclusion of figures, it would be more interesting to add graphics once we complete the experimental phases and proceed to publish the results.
It also would be good if Discussion section would be added, which would include the analysis of the presented information, and it would be quite useful to add the section Future Direction to demonstrate the thoughts about the questions which remained unanswered.
The authors have not added Discussion section as it is a review and do not include new experimental results, but Future directions section has been included before the Conclusions section.
Please check the following through the text – the dots at the ends of the paragraphs’ titles should be removed (lines 3, 61, 64, 94, 110, 132, 146, 156, 171, 188, 199, 204, 232, 244, 283, 313, 371, 414, 415, 474, 541, 567, 604, 622, 634, 646, 672, 696, 721).
The authors have removed all these dots as well as the one in the line 773.
Line 94 It would be better to locate the title on the same page with the paragraph content.
The authors have added information suggested by other reviewer and, for this reason, that title is now located on the next page (line 100).
In addition, the title “2.8. Sheep” is located now at line 196 to be at the same page of the paragraph content.
Line 158 LTT abbreviature should be deciphered.
The authors have explained the meaning of the abbreviature and changed the structure of the sentence for a better understanding.
Line 194 It would be better to decipher the following abbreviations: ANG and PGF.
These abbreviations have been explained as well as VEGF for a better understanding.
Line 233 It would be better to decipher the following abbreviation: ES.
The authors have not written the full name as it has been explained in the line 45 (embryonic stem, ES).
Line 241 The following abbreviation ‘iPS’ should be checked.
The abbreviation has been corrected: “iPSCs” instead of “iPS”.
Line 269 In the following phrase ‘in MSCs levels did not return to’ it would be better to specify levels of what.
The authors have changed the sentence to “and, in MSCs, ATM activation levels did not return to basal level by 48h”.
Line 313 It would be better to locate the title of the paragraph on the same page with its content.
The title is now located on the next page (line 322).
Line 348 It would be better to decipher the following abbreviations: ‘KGF, PDGF-A, PDGF-B, FGF-2, FGF-6, ANGPT-1, NGF, BDNF, GDNF, IGF-1, IL1RA’.
The authors have added the explanation of these abbreviations: “keratinocyte growth factor (KGF), platelet-derived growth factor subunit A (PDGFA), platelet-derived growth factor subunit B (PDGFB), fibroblast growth factor 2 (FGF2), fibroblast growth factor 6 (FGF6), Angiopoietin 1 (ANGPT1), nerve growth factor (NGF), brain derived neurotrophic factor (BDNF), glial cell line-derived neurotrophic factor (GDNF), insulin-like growth factor 1 (IGF-1), interleukin-1 receptor antagonist (IL1RA)”.
Line 349 It would be better to decipher the following abbreviations: ‘SCF, SDF-1’.
The abbreviations have been described as “stem cell factor (SCF)” and “stromal cell-derived factor 1 (SDF-1)”.
Line 350 It would be better to decipher the following abbreviation: ‘PGE2’.
The authors have not written the full name because it has been explained before (lines 358-359).
Line 351 It would be better to decipher the following abbreviation: ‘TSG-6’.
The authors have explained the meaning of the abbreviation: “tumor necrosis factor-α-stimulated gene/protein-6 (TSG-6)”.
Line 353 It would be better to decipher the following abbreviation: ‘TNF’.
The abbreviation has been described as “tumor necrosis factor (TNF)”.
Line 455 The following word ‘de’ should be checked.
The word “de” has been changed to “the”.
Line 610 The following fragment ‘1 x 106’ should be corrected.
The authors have changed “1 x 106” to “1 x 106”.
Line 717 It would be better to decipher the abbreviature ‘MMP9’.
The authors have not explained the meaning because it has been written before (line 350: “matrix metalloproteases-1 (MMP1), MMP2 and MMP9”).
Line 742 It would be better to start the following sentence ‘5 days following injection’ with the word.
The authors have changed the beginning of the sentence to “Five days following injection”.
Would you please check, the references should be described according the requirements published on the Journal’s website.
The authors have deleted the DOI of the articles according to the requirements published on the Journal’s website.
Additional modifications have been done for a better understanding of the work:
- Line 51.
- Lines 83-84.